# Impact of Regularization on Calibration and Robustness: from the Representation Space Perspective

## Abstract

Recent studies have shown that regularization techniques using soft labels, e.g., label smoothing, Mixup, and CutMix, not only enhance image classification accuracy but also improve model calibration and robustness against adversarial attacks. However, the underlying mechanisms of such improvements remain under-explored. In this paper, we offer a novel explanation from the perspective of the representation space (i.e., the space of the features obtained at the penultimate layer). Our investigation first reveals that the decision regions in the representation space form cone-like shapes around the origin after training regardless of the presence of regularization. However, applying regularization causes changes in the distribution of features (or representation vectors). The magnitudes of the representation vectors are reduced and subsequently the cosine similarities between the representation vectors and the class centers (minimal loss points for each class) become higher, which acts as a central mechanism inducing improved calibration and robustness. Our findings provide new insights into the characteristics of the high-dimensional representation space in relation to training and regularization using soft labels.

## 1 Introduction

The drive to improve the performance of classification models has led to the development of various regularization methods that use soft labels instead of one-hot encoded hard labels for classification targets. The regularization techniques such as label smoothing (Szegedy et al., 2016), Mixup (Zhang et al., 2018), and CutMix (Yun et al., 2019) have demonstrated significant success in improving classification accuracy across various benchmarks.

However, their impact goes beyond accuracy improvement. Studies have shown that these techniques contribute to better-calibrated models, aligning predicted probabilities more closely to actual accuracy (Guo et al., 2017; Müller et al., 2019). Furthermore, they have been shown to strengthen model robustness against gradient-based adversarial attacks, where subtle, imperceptible noise is added to input data to intentionally mislead models (Goodfellow et al., 2014; Yun et al., 2019; Fu et al., 2020; Zhang et al., 2021).

While the benefits of soft labels are evident, the underlying mechanisms by which they achieve these improvements remain largely unexplained. This is where our study comes in. In this paper, we offer a deeper understanding of how soft labels enhance model calibration and adversarial robustness by *examining the model's representation space*. Intuitively, data points that are correctly classified with lower confidence are located near decision boundaries, making them more vulnerable to small perturbations and reducing robustness (Hein et al., 2019; Kim et al., 2024). Therefore, when calibration and robustness are investigated, it is crucial to explore how decision boundaries are formed and how features, i.e., the outputs of the penultimate layer are distributed within the decision regions. To understand the characteristics of decision regions and boundaries, we begin by analyzing their shapes that can be observed in low-dimensional (2D and 3D) visualizable representation spaces. We then examine whether these characteristics persist in the original high-dimensional representation space. Based on the results, we study how the feature distribution in the representation space changes depending on the use of soft labels and how such changes can improve calibration and robustness.

Our work can be summarized as follows:

1. We show that **the decision regions form cone-like structures around the origin of the representation space**, and explain that this is because **logits are calculated as the dot product between features (representation vectors) and the weight vectors of the classification layer**. This structure is consistent in both low and high-dimensional representation spaces across various models and different training recipes (e.g., regularizations, weight initializations).

2. We describe the distribution of features in the representation space using two metrics: (1) the magnitude of the features, and (2) the cosine similarity of the features to the *class center*, the point in the representation space where the classification loss is minimal. Our findings demonstrate that **training models with regularization significantly reduces the magnitude of the features, leading to tighter clustering around the class center**.

3. Using the findings above, we explain why regularization using soft labels leads to improved calibration and robustness. We show that **feature vectors with smaller magnitudes improve model calibration**, as reducing the feature magnitude acts similarly to temperature scaling, a common post-hoc calibration method. Furthermore, by analyzing gradient directions in the representation space, we show that **smaller features tend to be distributed in robust regions**, which align better with the class center vector.

## 2    RELATED WORK

**Calibration and robustness.** Calibration refers to the alignment between a model's confidence and its actual accuracy. Guo et al. (2017) found that modern neural networks often exhibit overconfidence, leading to miscalibrated predictions. To address this, various techniques have been proposed, including temperature scaling (Guo et al., 2017), which is a single-parameter variant of Platt scaling (Platt et al., 1999). Concurrently, the vulnerability of neural networks to adversarial attacks has received significant attention since it was demonstrated that imperceptible input perturbations could lead to misclassifications (Szegedy et al., 2013). Moreover, the introduction of the Fast Gradient Sign Method (FGSM) (Goodfellow et al., 2014) has spurred the development of numerous attack and defense strategies (Carlini & Wagner, 2017; Madry et al., 2018; Croce & Hein, 2020; Deng & Mu, 2024).

**Regularization techniques.** Data augmentation and regularization techniques such as label smoothing, Mixup, and CutMix have gained significant attention for their ability to improve model generalization. Label smoothing distributes a small portion of probability mass uniformly across all labels, softening the one-hot encoded targets (Szegedy et al., 2016). Mixup linearly interpolates both inputs and labels, generating virtual training examples (Zhang et al., 2018). CutMix extends this idea by replacing rectangular regions in one image with patches from another, adjusting labels proportionally (Yun et al., 2019). These methods have demonstrated promising results not only in enhancing model generalization but also in improving model calibration and robustness to adversarial attacks (Yun et al., 2019; Fu et al., 2020; Zhang et al., 2021).

There are studies investigating why these regularization techniques enhance model performance. Regarding calibration, Thulasidasan et al. (2019) examined whether the improvement in calibration is due to augmented data preventing memorization and overfitting. They trained models with convex combinations of images but used hard labels. The results indicated that simply mixing features does not improve calibration, emphasizing that smooth labels are crucial for achieving well-calibrated models. Recently, visualizations have shown that Mixup-generated training data tend to cluster near decision boundaries, leading the models to make less confident predictions and reducing miscalibration caused by overconfidence (Fisher et al., 2024). In terms of robustness to adversarial attacks, Zhang et al. (2021) demonstrated that minimizing the Mixup loss is approximately equivalent to minimizing an upper bound of the adversarial loss, thereby improving robustness. However, there is still a lack of a comprehensive explanation for why soft labels enhance calibration and robustness from the perspective of representation space, which is the focus of this paper.

**Representation space.** Several studies have explored the dynamics of the representation space in deep learning models. Wang et al. (2017) visualized the representation space using 2-dimensional features and found that features are distributed in a radial pattern. Another notable concept is Neural Collapse, which shows that both weight vectors and feature vectors converge to an Equiangular

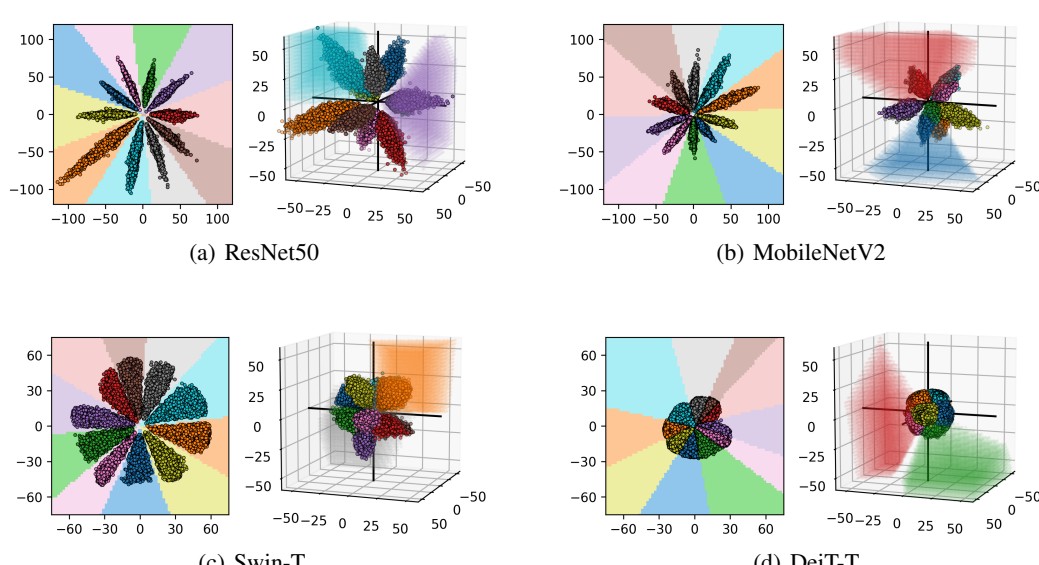

Figure 1: Visualization of 2D and 3D representation spaces of ResNet50, MobileNetV2, Swin-T, and DeiT-T on CIFAR-10. Circled dots represent output features, with different colors indicating different classes. The whole 2D planes are also colored according to the classification result of each point in the plane. In the case of 3D, regions corresponding only two classes are colored as examples for the sake of visualization. The values of the feature vectors are used as coordinates for the x, y, and z axes. Note that the scales differ across figures to best visualize the representation spaces.

Tight Frame (ETF) structure (Papyan et al., 2020). Extending this idea to the field of adversarial attacks, Su et al. (2024) showed that the angular distance between the feature means of clean images and perturbed images is generally small. Regarding the effect of regularization on the representation space, it was observed that label smoothing tends to bring features closer together (Müller et al., 2019). However, no studies have explained why such representations improve calibration and robustness, and how decision boundaries are formed to distinguish representations from different classes, which are addressed in this paper. Furthermore, the aforementioned studies primarily focus on convolutional neural networks (CNNs); we extend the investigation to a broader range of model types, including vision transformers (ViTs).

## 3 DECISION REGIONS IN THE REPRESENTATION SPACE

As mentioned in the introduction, when evaluating calibration and robustness, it is necessary to examine how decision boundaries are formed and how features are distributed within the decision regions. In this section, we visualize the representation space to show how the decision regions are shaped and explain the reasons behind these shapes.

### 3.1 2D AND 3D REPRESENTATION SPACES

A classification model typically consists of a feature extractor that maps inputs into features and a classification layer that uses those features to make decisions. The representation space of the model refers to the space where the output of the feature extractor, or more specifically, the output of the penultimate layer of the model resides. It usually has high dimensionality (e.g., 2048 for ResNet50 and 768 for Swin-T), making it challenging to visually analyze its characteristics. To address this, we insert a linear layer between the feature extractor and the classification layer, mapping the output features into 2D (or 3D) vectors, and adjust the input dimension of the classification layer accordingly. As a result, the new representation space is 2D (or 3D), which facilitates visual examination.

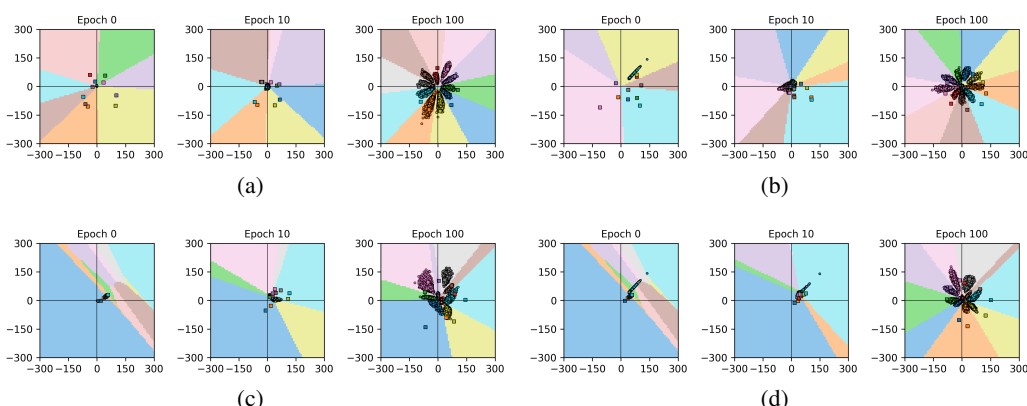

Figure 2: Changes in the 2D representation space as training progresses for ResNet50 on CIFAR-10 with different weight initializations. Features and weight vectors are represented as circles and squares, respectively (weight vectors are scaled to indicate direction).

We train this modified model as described in Appendix A.1 and visualize the 2D (or 3D) representation space as described in Appendix B.

Figure 1 shows the visualization results for ResNet50 (He et al., 2016), MobileNetV2 (Howard, 2017), Swin-T (Liu et al., 2021), and DeiT-T (Touvron et al., 2021) on the CIFAR-10 dataset (Krizhevsky et al., 2009). Regardless of the differences in model structures, it can be observed that the decision regions are divided into circular sectors, i.e., *cone-like shapes*, centered around the origin, with features radially distributed within these regions.

In addition, these characteristics remain consistent across different weight initializations as well. Figure 2 shows the process of how the decision regions change during training of ResNet50 on CIFAR-10 with four different weight initializations (see Appendix A.1 for details). In Figure 2(a), we use the default Kaiming uniform weight initialization (He et al., 2015). Before training, the features are distributed close to the origin, as the model's weights are initialized as small values, and some decision regions are divided around it. For other decision regions, due to random weight initialization, they have not yet been assigned to regions in the representation space. In Figure 2(b), we set the initial weights of the feature extractor in such a way that the features are positioned far from the origin while the weights of the classification layer are initialized as in Figure 2(a) so that the decision regions remain divided around the origin. In Figure 2(c), we set the initial weights of the classification layer so that the decision regions are divided far from the origin (but the initial features are located around the origin due to the default initialization of the feature extractor). Finally, in Figure 2(d), we adjust the initial weights of the entire model so that shifting the center of the decision regions is shifted and the features are distributed far from the origin. It can be observed from the figure that in all four cases, the decision regions eventually evolve into cone-like shapes around the origin as training progresses.

The cone-shaped decision regions are also consistently observed when regularization using soft labels is applied (see Figure 10 in Appendix D).

## 3.2 ORIGINAL REPRESENTATION SPACES

In this section, we demonstrate that decision regions in the original, high-dimensional representation space (e.g., 2048 for ResNet50) are also divided into cone shapes centered around the origin. One simple way to verify this is to gradually move a correctly classified feature linearly toward the origin and observe when it becomes misclassified for the first time. This process is illustrated in Figure 3. If the decision regions are cone-shaped, the classification result will remain consistent until the feature arrives at the origin. Actually, the intersection point of the cone-shaped decision regions does not precisely coincide with the origin, but is close to the origin. Thus, the moment that the misclassification occurs will be only at the final stage of the linear movement. On the other hand,

if the regions are not cone-shaped, meaning another class region lies between the feature and the origin, the feature will become misclassified early during the movement.

We verify this for ResNet50, MobileNetV2, Swin-T, and DeiT-T on the test sets of CIFAR-10 and CIFAR-100 (Krizhevsky et al., 2009), and the validation set of ImageNet (Russakovsky et al., 2015) (see Appendix A.2 for training details). For each feature in the representation space, we linearly move it toward the origin over 100 uniform steps. If the index of the first misclassified step is close to 100, it suggests that the decision region is likely cone-shaped. The results are shown in Table 1. Since nearly all indices are over 98, we can confirm that decision regions are indeed divided into cone shapes in high-dimensional representation spaces as well. Further verification can be found in Appendix C.

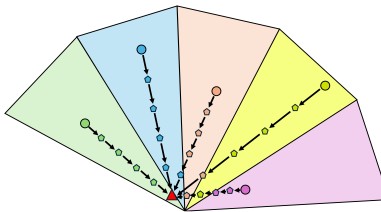

Table 1: Accuracy and the mean (and the standard deviation) of the first movement index of misclassification for various models on different datasets.

| Model | Dataset | Accuracy (%) | Index |
|---|---|---|---|
| ResNet50 | CIFAR-10 | 92.5 | 99.9 ±1.6 |
| | CIFAR-100 | 71.4 | 99.4 ±3.9 |
| | ImageNet | 76.1 | 99.8 ±1.9 |
| MobileNetV2 | CIFAR-10 | 92.6 | 99.8 ±1.7 |
| | CIFAR-100 | 71.7 | 98.9 ±4.2 |
| | ImageNet | 71.9 | 98.5 ±4.9 |
| Swin-T | CIFAR-10 | 89.3 | 99.9 ±1.3 |
| | CIFAR-100 | 66.7 | 99.6 ±2.7 |
| | ImageNet | 75.8 | 99.3 ±3.2 |
| DeiT-T | CIFAR-10 | 81.2 | 99.7 ±2.6 |
| | CIFAR-100 | 50.3 | 98.2 ±5.5 |
| | ImageNet | 72.0 | 90.0 ±9.0 |

Figure 3: Illustration of linear movements of features (circled dots) toward the origin (red triangle). Each large triangular region represents the decision region of a specific class. Pentagons represent the intermediate positions of features as they move toward the origin. The colors within each dot indicate the class to which they are classified.

### 3.3 WHY ARE DECISION REGIONS CONE-SHAPED?

To understand the shape of the decision regions, we investigate what happens when a feature passes through the classification layer. In other words, we examine the factors that influence the calculation of logits. A feature is assigned to the class showing the largest logit value by the classification layer. Given a feature $\boldsymbol{f}$ for an input image, the logit $y_c$ for class $c$ can be expressed as follows.

$$y_c = \boldsymbol{w}_c^T \boldsymbol{f} + b_c = ||\boldsymbol{w}_c|| \cdot ||\boldsymbol{f}|| \cos \theta + b_c, \tag{1}$$

where $\boldsymbol{w}_c$ and $b_c$ are the weight vector and the bias of the classification layer for class $c$, respectively, and $\theta$ is the angle between $\boldsymbol{f}$ and $\boldsymbol{w}_c$. Therefore, the elements that can affect the prediction result for an image are $||\boldsymbol{w}_c||$, $\cos \theta$, and $b_c$.

Note that $||\boldsymbol{w}_c||$ and $b_c$ are commonly applied to all data, so they are trained not to vary much across different classes (Papyan et al., 2020). In addition, we find that most bias values are very close to zero at the end of training and thus have minimal impact on the ranking of logit values (further details are provided in Appendix C). The only remaining factor that can play a decisive role in the classification result is the cosine similarity between the feature vector $\boldsymbol{f}$ and the weight vector $\boldsymbol{w}_c$ for class $c$. In other words, the classification result depends on the alignment between the class's weight vector and the feature vector. Thus, decision boundaries are formed based on the degree of alignment, and consequently, the decision regions take the shape of cones centered at the origin, aligned with the weight vectors. This explains why different model structures (differing only in feature extractors, while using the same classification layers as in Equation 1) and varying weight initializations do not alter the characteristic cone-shaped decision regions forming around the origin.

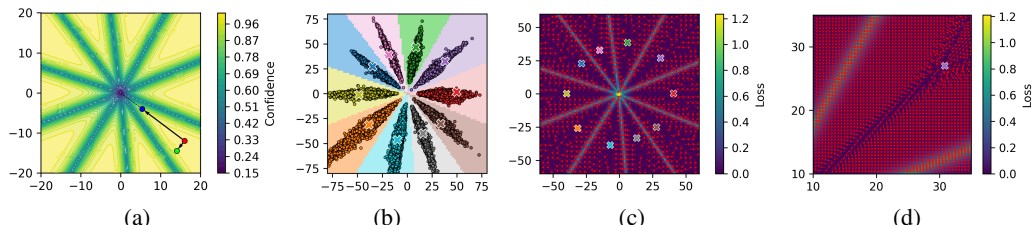

(a)       (b)       (c)       (d)

Figure 4: 2D representation space of ResNet50 on CIFAR-10 as in Figure 1(a). Each cross mark represents the class center, i.e., the location with the lowest loss for each class. (a) Confidence contours. (b) Decision regions and feature distributions. (c) Loss and gradient directions. (d) Enlarged version of (c).

## 4 EFFECT OF REGULARIZATION

In Section 3, we demonstrated that decision regions form cone-like shapes around the origin in the representation space. In this section, we investigate the distribution of features within this space. Specifically, we describe our methods for determining how far a feature is located from the decision boundary and explain how features are distributed based on these methods. We then discuss how the feature distribution changes due to regularization and how these changes are related to improvements in calibration and robustness performance.

### 4.1 ANALYSIS METHODS

In a low-dimensional space, it is easy to identify the approximate distribution of features, as shown in Figure 1. In a high-dimensional space, however, visualization becomes challenging, making it difficult to understand how features are distributed. To overcome this, we establish criteria for analysis of feature distributions. In particular, we quantify how close a feature is to the decision boundary based on its confidence value.

In Figure 4(a), we present the confidence contours of decision regions in the 2D representation space. Suppose that we want to move a correctly classified feature with high confidence (red dot) to reduce its confidence. This can occur in two ways: 1) by moving radially toward the origin (blue dot) or 2) by moving toward the nearest decision boundary (green dot). Therefore, we determine how far a specific feature is from the decision boundary using two criteria: 1) **the root mean square (RMS) of the feature** (we use RMS as the feature magnitude to compensate for different dimensionalities across models) and 2) **the cosine similarity of the feature with the *class center*** (crosses in Figure 4(b)), where the classification loss is minimal for that class. The class center for a certain class is found by optimizing an arbitrary feature vector in the representation space to achieve the lowest loss for that class using the gradient descent method. Figure 4(c) shows the loss and gradient directions in the 2D representation space, where the point from which the gradients flow out corresponds to the class center. Discussions on different candidates for the class center can be found in Appendix E, where we verify that the minimum loss point is the best option for measuring proximity to the decision boundary based on confidence.

We show the relationship of the confidence vs. the RMS of features and the cosine similarity of features with the class center in the top row of Figure 5. As expected, the smaller the RMS of features or the lower the cosine similarity is, the lower the confidence is, indicating proximity to the decision boundary.

### 4.2 IMPACT ON FEATURE DISTRIBUTION

In Figure 6, we present the results of training with and without regularization (label smoothing, Mixup, and CutMix) in the 2D representation space. Training with regularization results in two notable changes. First, the RMS of features significantly decreases (top row of Figure 5 and Figure 6), bringing them closer to the origin. Second, from the middle row of Figure 5, we observe that the

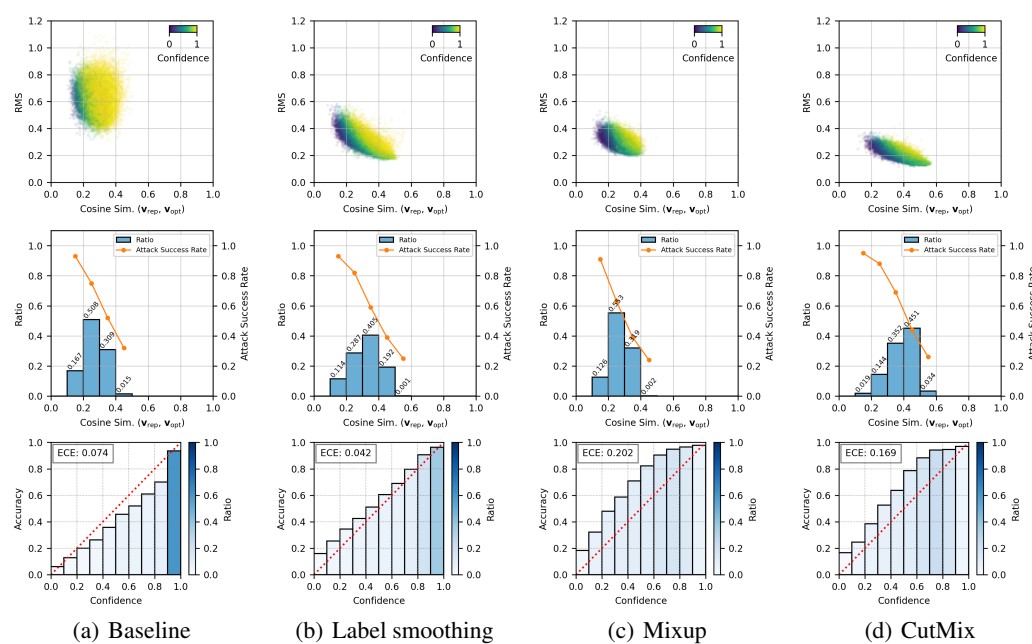

(a) Baseline  (b) Label smoothing  (c) Mixup  (d) CutMix

Figure 5: Evaluation results of ResNet50 on the ImageNet validation data. **Top.** Scatter plots of feature RMS and cosine similarities of features ($\mathbf{v}_{rep}$) with the class center ($\mathbf{v}_{opt}$). Colors represent confidence values. **Middle.** Histograms of cosine similarities of features to the class center, along with the FGSM attack success rate for each bin. **Bottom.** Reliability diagrams, where the transparency of bars represents the ratio of data in each confidence bin. ECE values are shown for each case.

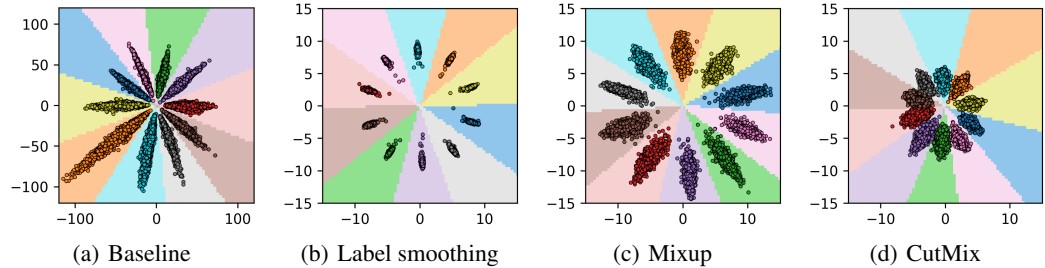

(a) Baseline  (b) Label smoothing  (c) Mixup  (d) CutMix

Figure 6: Features in the 2D representation space for different regularization methods for ResNet50 on CIFAR-10. Note that the scales differ across figures.

proportion of data with high cosine similarity between the feature and the class center increases in the regularized models. These changes are examined in detail below.

**Decrease in RMS.** When one-hot encoded labels are used for training, the weights are updated to maximize the difference between logits. To achieve great logit separation between classes, it is beneficial to have large feature vectors (Wang et al., 2017). However, with soft labels, the deviation between logit values should be reduced, as large feature vectors would increase the loss. Therefore, a model trained with hard labels learns to generate features farther from the origin (with large RMS), while a model trained with soft labels produce features closer to the origin (with small RMS). To confirm this, we visualize the cross-entropy loss and gradient directions for hard and soft labels in a 2D representation space in Figures 7(a) and 7(b), respectively. The point with the smallest loss is marked with a white cross. We can see that for hard labels, the location of the cross mark is far from the origin, while for soft labels, it is located near the origin.

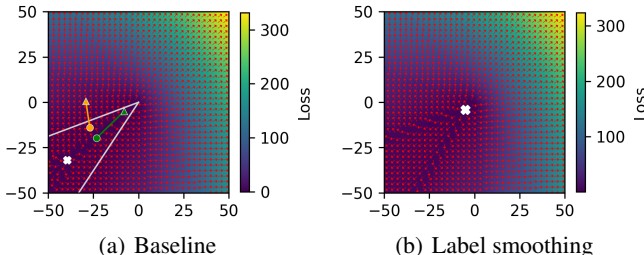

(a) Baseline          (b) Label smoothing

Figure 7: Loss and gradient directions for a certain class in the 2D representation space of ResNet50 on CIFAR-10. White crosses indicate the location with the smallest loss. Circles and triangles represent the features of clean and perturbed data, respectively. White lines depict the decision boundary. (a) Trained without regularization. (b) Trained with regularization (label smoothing).

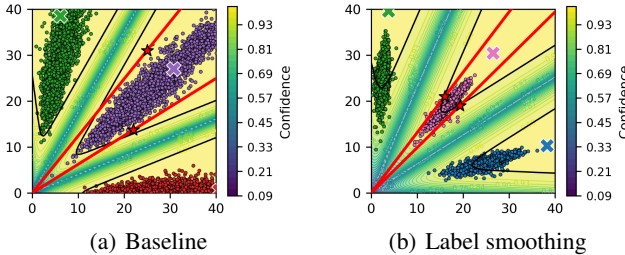

(a) Baseline          (b) Label smoothing

Figure 8: Confidence contours and features (circled dots) in the 2D representation space. The 0.99 confidence contour is shown as a black line. Crosses indicate the minimum loss points. Red stars represent the features with confidence higher than 0.99 but with the poorest alignment to their class center in terms of cosine similarity; red lines connect them with the origin.

**Increase in cosine similarity.** In Figure 8, we show the confidence contours in the 2D representation space of ResNet50 trained with and without regularization on CIFAR-10. For both models, we compare how well a feature needs to be aligned with the class center (crosses) to achieve a certain confidence level. Specifically, we search for features with confidence higher than 0.99 but with the worst alignment to their class center in terms of cosine similarity, in both clockwise and counterclockwise directions (red stars). By connecting these features to the origin (red lines) and observing the angle between the lines, we can see that the angle in the regularized model is smaller (Figure 8(b)). This occurs because, near the origin, the region for achieving a certain confidence value (e.g., 0.99) or higher becomes narrow. Since this region contains the class center, a feature close to the origin must be well-aligned with the class center to reach a given confidence level. Conversely, a feature located far from the origin can still achieve high confidence without being as closely aligned with the class center as a feature located near the origin (Figure 8(a)). Therefore, in the regularized models producing features with small RMS, the cosine similarity of features with the class center is relatively high compared to the baseline models.

### 4.3 IMPACT ON CALIBRATION

In the bottom row of Figure 5, miscalibration due to overconfidence in the baseline training is suppressed using regularization. This can be explained in relation to the RMS decrease mentioned in Section 4.2.

When the magnitude of a feature $\boldsymbol{f}$ is decreased (red dot becoming the blue dot in Figure 4(a)) by a factor of $T$ due to training with regularization, its corresponding logit for class $c$ can be expressed as $\frac{\boldsymbol{w}_c^T \boldsymbol{f}}{T} + b_c$. In Section 3, we demonstrated that, due to the cone-shaped decision boundaries, vectors located on the line connecting a feature and the origin are mostly classified into the same class as the feature (Table 1). Furthermore, in Section 4.1, we showed that features with smaller RMS have lower confidence values. Therefore, if the magnitude of a feature vector is scaled down and the

feature moves closer to the origin, the confidence of the feature will decrease, but the prediction will remain unchanged. In other words, by scaling the features through regularization using soft labels, the model's calibration is adjusted without affecting accuracy. See Appendix G for additional related experiments.

In fact, the effect of feature scaling due to regularization is similar to a post-processing technique known as temperature scaling. Temperature scaling adjusts calibration by scaling the logit values by a factor of $T$, resulting in the logit expression $\frac{\boldsymbol{w}_c^T \boldsymbol{f} + b_c}{T}$, which is similar to the case of feature scaling. Mathematically, there is a difference of $\frac{T-1}{T} b_c$, but as discussed in Section 3.3, most bias values are close to zero and do not affect the ranking of logits.

## 4.4 Impact on Adversarial Robustness

How does the use of regularization lead to better robustness against gradient-based adversarial attacks? To explain this, we examine the gradient directions in the 2D space of ResNet50 on CIFAR-10 in Figure 7(a). Note that the gradients shown in the figure is used to perturb the data in FGSM (and iterative FGSM), which is the most basic gradient-based attack. In areas with high cosine similarity to the class center (marked with a cross), the gradient directions point toward the origin. However, in areas with low cosine similarity, the gradients are nearly orthogonal to the class center vector, heading toward the decision boundary. Specifically, we show two samples in Figure 7(a): circles represent the features of correctly classified clean images, and triangles represent features after perturbation by an amount of $\epsilon = 8/255$. The feature vector well-aligned with the class center vector (green sample) can remain within the decision region after being perturbed, as the gradient points toward the origin. On the other hand, the feature vector poorly aligned with the class center vector (orange sample) moves toward the nearby decision boundary and easily becomes misclassified.

To verify this in the original high-dimensional representation space, we apply the same attack to ResNet50 on ImageNet. The results are shown in the middle row of Figure 5, where the blue bars represent the histogram of cosine similarity between features and their class centers, and the orange line shows the attack success rate for each confidence bin. For regularized models, the number of features with high cosine similarity to the class center increases, and these features exhibit lower attack success rates, which is consistent with the results observed in the 2D representation space. Further discussions on adversarial robustness are provided in Appendix H.

## 4.5 Comprehensive Evaluation

In Table 2, we present comprehensive results for various models (ResNet50, Swin-T, MobileNetV2 (Howard, 2017), EfficientNet-B1 (Tan, 2019), and ViT-B/16 (Dosovitskiy et al., 2021)) trained with different methods on the ImageNet dataset (see Appendix A.2 for training details). We consistently observe that, when regularization is applied, the RMS of features decreases, and the cosine similarity between features and class centers increases. These changes result in reduced overconfidence in predictions (leading even to underconfidence in some cases) and improved robustness to adversarial attacks.

## 5 Discussion

There exist numerous studies that compare the calibration and adversarial robustness performances between convolution-based and transformer-based models. For calibration, results indicate that transformer-based models are better calibrated than convolutional models (Minderer et al., 2021). Regarding adversarial robustness, models adopting the transformer architectures have been shown to be more robust to adversarial perturbations (Bai et al., 2021; Benz et al., 2021; Paul & Chen, 2022).

However, we argue that the effect of soft label-based regularization has been often overlooked in such comparisons. Transformer models typically employ various regularization techniques during pretraining and training phases, such as label smoothing, Mixup, and CutMix. In contrast, convolutional models are often evaluated without such extensive regularization techniques, as they can achieve reasonable test accuracies without relying heavily on these methods. This inconsistency in the application of regularization techniques may introduce a bias in the comparison results, poten-

Table 2: Overall performance and feature statistics (mean and standard deviation values) across various models and training methods. Red ECE values indicate overconfidence, while blue ECE values indicate underconfidence. For MobileNetV2, EfficientNet-B1, and ViT-B/16, we present the results using pretrained weights from PyTorch, where the second row in each model corresponds to stronger regularization (see Table 3 in Appendix A.2).

| Model | Method | Validation Accuracy | RMS | Cosine Similarity | ECE | Attack Success Rate |
|---|---|---|---|---|---|---|
| ResNet50 | Baseline | 76.1 | 0.62 ±0.09 | 0.27 ±0.06 | 0.074 | 67.2 |
| | Label smoothing | 77.1 | 0.29 ±0.07 | 0.32 ±0.09 | 0.042 | 61.6 |
| | Mixup | 76.6 | 0.30 ±0.04 | 0.27 ±0.05 | 0.202 | 55.0 |
| | CutMix | 78.0 | 0.20 ±0.04 | 0.38 ±0.08 | 0.169 | 55.2 |
| Swin-T | Baseline | 75.8 | 1.38 ±0.06 | 0.36 ±0.09 | 0.095 | 83.9 |
| | Label smoothing | 76.3 | 0.67 ±0.14 | 0.44 ±0.13 | 0.033 | 79.0 |
| | Mixup | 78.2 | 0.66 ±0.13 | 0.46 ±0.11 | 0.013 | 71.9 |
| | CutMix | 78.7 | 0.72 ±0.13 | 0.51 ±0.13 | 0.050 | 77.1 |
| MobileNetV2 | PyTorch V1 | 71.9 | 0.78 ±0.09 | 0.31 ±0.07 | 0.028 | 85.7 |
| | PyTorch V2 | 72.0 | 0.28 ±0.05 | 0.36 ±0.09 | 0.367 | 73.2 |
| EfficientNet-B1 | PyTorch V1 | 77.6 | 0.34 ±0.08 | 0.32 ±0.09 | 0.091 | 65.1 |
| | PyTorch V2 | 78.9 | 0.15 ±0.02 | 0.34 ±0.07 | 0.271 | 62.1 |
| ViT-B/16 | PyTorch Swag Linear V1 | 81.8 | 1.28 ±0.08 | 0.33 ±0.08 | 0.018 | 58.8 |
| | PyTorch V1 | 81.1 | 0.56 ±0.09 | 0.57 ±0.11 | 0.055 | 54.9 |

tially overstating the advantages of transformer architectures over convolutional models. To conduct a more equitable comparison, it is crucial to consider and account for these differences in regularization strategies between the two model families.

In Table 2, we present results of two models (ResNet50 and Swin-T) that we trained on the ImageNet dataset. ResNet50, a convolutional-based model, has a similar number of parameters (25.5M) to Swin-T, a transformer-based model (28.2M). Using identical training recipes (see Appendix A.2), they achieve comparable validation accuracy. Under these equitable training conditions, unlike prior studies, we find that Swin-T is as overconfident as ResNet50 when no regularization is applied. In addition, we observe that Swin-T is actually more vulnerable to adversarial attack in all cases (baseline, label smoothing, Mixup, and CutMix). Although the overall cosine similarity between features and class centers of Swin-T is higher than that of ResNet50, it should be noted that the cosine similarity may be limited to compare different models due to the difference in the dimension of the representation space (2048 in ResNet50 and 768 in Swin-T).

## 6 CONCLUSION

In this paper, we have explored the underlying mechanisms through which regularization techniques using soft labels, such as label smoothing, Mixup, and CutMix, enhance both model calibration and robustness to gradient-based adversarial attacks. Our investigation focused on how decision regions are formed and how regularization influences feature distributions in the representation space.

Our analysis revealed that decision regions form a cone-like structure around the origin, with features distributed radially within these boundaries. Additionally, we showed that regularization reduces the RMS of representation vectors, leading to tighter clustering of them. We further explained that the formation of tighter clusters in regions with small RMS not only improves calibration by mimicking the effect of temperature scaling but also increases resilience to adversarial perturbations.

We believe that these findings provide a new perspective on the dynamics induced by regularization in the representation space. Nevertheless, our study also calls for follow-up studies in several directions. In particular, we plan to extend our analysis to investigate dependence on model components and regularization hyperparameters.

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

# A   IMPLEMENTATION DETAILS

## A.1   2D AND 3D REPRESENTATION SPACES

We train models on the CIFAR-10 dataset for 100 epochs using the SGD optimizer with a momentum of 0.9 and a weight decay of 0.0001. For learning rate scheduling, we apply a linear warmup from 0 to 0.01 over the first 10 epochs, followed by a cosine annealing scheduler for the remaining 90 epochs. Regarding regularization hyperparameters, we set the label smoothing value to 0.1 and use an alpha value of 0.2 for both Mixup and CutMix.

Regarding weight initialization in Figure 2, to initialize the model to distribute features far from the origin, we first train the feature extractor alone, using the L2 norm between the extracted features and $\mathbf{v} = [80, 80]$ as the loss function for 1 epoch using the SGD optimizer with a learning rate of 0.0005. After this, we attach a linear classifier and train the full model using the training settings listed above. To initialize the classifier so that the decision regions are divided far from the origin, we use hand-crafted parameters, where the weight matrix $\boldsymbol{W}$ and bias vector $\boldsymbol{b}$ are defined as follows:

$$\boldsymbol{W} = \begin{bmatrix} 0.1318 & 0.2245 & 0.2630 & 0.2881 & 0.2947 & 0.3189 & 0.3195 & 0.3323 & 0.3482 & 0.3619 \\ -0.0165 & 0.0709 & 0.1030 & 0.1138 & 0.1381 & 0.1362 & 0.1567 & 0.1654 & 0.1684 & 0.1806 \end{bmatrix},$$

$$\boldsymbol{b} = \begin{bmatrix} 15.705 \\ 9.045 \\ 5.644 \\ 3.144 \\ 0.834 \\ -1.286 \\ -3.442 \\ -5.799 \\ -8.489 \\ -11.972 \end{bmatrix}.$$

Note that for the results shown in Figure 2, regularization using soft labels is not applied.

## A.2   ORIGINAL REPRESENTATION SPACES

We train models on the CIFAR-10, CIFAR-100, and ImageNet datasets for 300 epochs using the AdamW optimizer (Loshchilov, 2017) with a weight decay of 0.0005. For learning rate scheduling, we apply a linear warmup from 0 to 0.001 over the first 20 epochs, followed by a cosine annealing scheduler for the remaining 280 epochs. When training Swin-T on CIFAR-10 and CIFAR-100, we increase the learning rate from 0 to 0.01 during the warmup phase to improve test accuracy. Regarding the regularization hyperparameters, we set the label smoothing value to 0.1 and use an alpha value of 1.0 for both Mixup and CutMix.

The weights used for MobileNetV2 and DeiT-T in Table 1, as well as for MobileNetV2, EfficientNet-B1, and ViT-B/16 in Tables 2, 4, and 5, are pretrained weights from PyTorch. In the PyTorch framework, these pretrained weights can be accessed using the names `IMAGENET1K_V1`, `IMAGENET1K_V2`, and `IMAGENET1K_SWAG_LINEAR_V1`. Details on the regularization hyperparameters (using soft labels) used to train these weights can be found in Table 3.

Table 3: Regularization hyperparameters to train models on ImageNet.

| Model | Method | Label Smoothing | Mixup | CutMix |
|---|---|---|---|---|
| MobileNetV2 | PyTorch V1 | - | - | - |
|  | PyTorch V2 | 0.10 | 0.2 | 1.0 |
| EfficientNet-B1 | PyTorch V1 | - | - | - |
|  | PyTorch V2 | 0.10 | 0.2 | 1.0 |
| ViT-B/16 | PyTorch Swag Linear V1 | - | 0.1 | - |
|  | PyTorch V1 | 0.11 | 0.2 | 1.0 |

## B    REPRESENTATION SPACE VISUALIZATION

To distinguish between different decision regions in the 2D (or 3D) representation space, we input a 2D grid (or 3D cube) with a fixed range into the classification layer. Each point in the grid or cube is then classified into a specific class. We visualize the decision regions by coloring each point in the grid or cube according to its predicted class. Next, using the values of the 2D (or 3D) feature vectors as coordinates, we plot their locations in the representation space, marking them with black-bordered circles colored by their predicted class. This process allows us to visualize how decision regions are divided and how features are distributed within these regions, as in Figures 1, 2, 4(b), 6, 9(a), 10, 11(a).

## C    BIAS TERM IN THE CLASSIFICATION LAYER

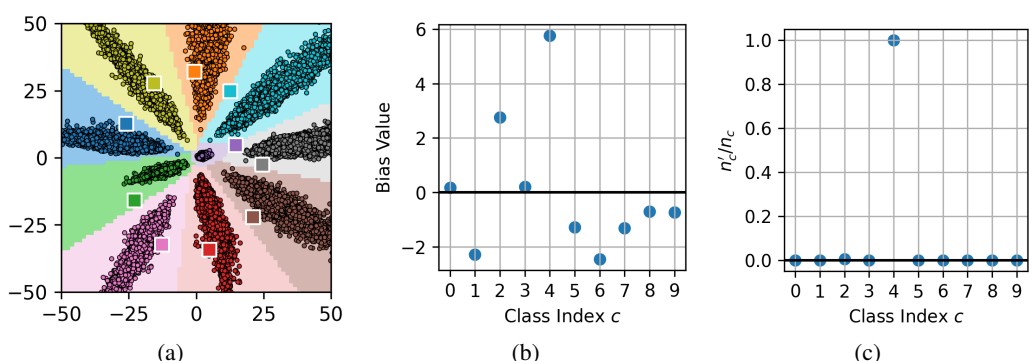

(a)                          (b)                          (c)

Figure 9: Results for ResNet50 with a 2D representation space trained on CIFAR-10. (a) 2D representation space. Circled and squared dots represent the features and weight vectors, respectively. Different colors indicate different class regions and classification results. (b) Bias values for each class. (c) $n'_c/n_c$ values for each class.

It is possible to obtain an imperfect structure of cone-shaped decision regions. An example is shown in Figure 9(a), which shows the 2D representation space of ResNet50 trained on CIFAR-10. While most classes form cone-shaped decision regions, a purple class with a circular decision region appears near the origin. This reflects the case described by Wang et al. (2017), where a class is determined by the bias value. As discussed in Section 3.3, cone-shaped decision regions arise because the classification result is determined by the weight vector that is aligned most closely with the feature vector. However, when two weight vectors have similar directions, as the purple and gray classes in Figure 9(a), the final classification is determined by the biases. To validate this, we examine the bias values for all 10 classes in Figure 9(b). It is clear that the purple class (class 4) has a significantly higher bias value compared to the other classes. This suggests that without the bias, features from class 4 would not be correctly classified.

We calculate the ratio of instances where prediction results depend on the bias values when determining logits. To elaborate, for an arbitrary class $c$, we count the number of correctly classified samples $n_c = \sum_{i=1}^{N} \mathbf{1}(\hat{y}_i = y_i = c)$, where $\hat{y}_i$ is the predicted class, $y_i$ is the true class for sample $i$, and $N$ is the total number of samples.

Then, let $\mathbb{A}_c$ be the set of indices of samples that are correctly predicted into class $c$. Among such samples, we count the number of samples $n'_c$ where the prediction results would change if the logits are calculated without biases. This can be expressed as $n'_c = \sum_{j \in \mathbb{A}_c} \mathbf{1}(\hat{y}_j^{\text{no bias}} \neq \hat{y}_j)$, where $\hat{y}_j^{\text{no bias}}$ is the predicted class for sample $j$ when logits are calculated without biases. Therefore, if the ratio $n'_c/n_c$ is large, the presence of bias values are crucial for correctly classifying data into class $c$.

Figure 9(c) shows the $n'_c/n_c$ values for each class. It is clear that classes with cone-shaped decision regions have low $n'_c/n_c$ values, indicating that bias terms are not necessary for correctly classifying

these classes. However, for class 4 (the purple class in Figure 9(a)), the value of $n'_c/n_c$ is 1, suggesting that non-cone-shaped decision regions rely on the bias for accurate classification. Therefore, by examining the $n'_c/n_c$ values, we can determine whether the bias is needed for correct classification of a particular class and infer the shape of its decision region in the representation space.

However, this bias-dependence phenomenon rarely occurs in the original high-dimensional representation spaces. Table 4 shows the mean, standard deviation, and maximum of $|b_c|$ (the absolute bias values for each class $c$) and the ratios $n'_c/n_c$ for models trained on ImageNet. The low $|b_c|$ values indicate minimal dependency on the bias for classification results, leading to smaller $n'_c/n_c$ values. Consequently, we obtain cone-shaped decision regions for all classes.

Why, then, did a non-cone-shaped decision boundary appear in Figure 9? This is likely due to the difficulty that the model faces when trying to fit multiple classes into a cone-like structure within a constrained dimensionality, such as fitting 10 classes (or even 100 classes for CIFAR-100) into a 2D space. More examples on the effect of the bias in the 2D representation space are provided in Appendix D.

Table 4: Mean, standard deviation, and maximum values of $|b_c|$ and $n'_c/n_c$ for models trained on ImageNet. Training details can be found in Appendix A.2.

| Model | Method | $|b_c|$ | | $n'_c/n_c$ | |
| --- | --- | --- | --- | --- | --- |
| | | Mean (±std) | Max | Mean (±std) | Max |
| ResNet50 | Baseline | 0.009 (±0.007) | 0.034 | 0.0005 (±0.005) | 0.10 |
| | Label smoothing | 0.011 (±0.008) | 0.048 | 0.0007 (±0.005) | 0.05 |
| | Mixup | 0.008 (±0.006) | 0.031 | 0.0005 (±0.005) | 0.08 |
| | CutMix | 0.009 (±0.007) | 0.047 | 0.0003 (±0.003) | 0.06 |
| Swin-T | Baseline | 0.029 (±0.022) | 0.138 | 0.0010 (±0.006) | 0.08 |
| | Label smoothing | 0.015 (±0.010) | 0.056 | 0.0009 (±0.006) | 0.09 |
| | Mixup | 0.026 (±0.020) | 0.106 | 0.0015 (±0.008) | 0.10 |
| | CutMix | 0.026 (±0.021) | 0.110 | 0.0013 (±0.007) | 0.07 |
| MobileNetV2 | PyTorch V1 | 0.028 (±0.022) | 0.155 | 0.0037 (±0.014) | 0.15 |
| | PyTorch V2 | 0.053 (±0.042) | 0.287 | 0.0084 (±0.023) | 0.25 |
| EfficientNet-B1 | PyTorch V1 | 0.054 (±0.041) | 0.263 | 0.0037 (±0.015) | 0.29 |
| | PyTorch V2 | 0.116 (±0.084) | 0.457 | 0.0065 (±0.021) | 0.23 |
| ViT-B/16 | PyTorch Swag Linear V1 | 0.030 (±0.026) | 0.226 | 0.0022 (±0.011) | 0.12 |
| | PyTorch V1 | 0.016 (±0.013) | 0.072 | 0.0007 (±0.005) | 0.08 |

# D  MORE VISUALIZATION ON 2D REPRESENTATION SPACE

Figure 10 provides visualization of decision regions on the 2D representation space. Considering the discussion in Section C, the results both with and without the bias terms in the classification layer are shown.

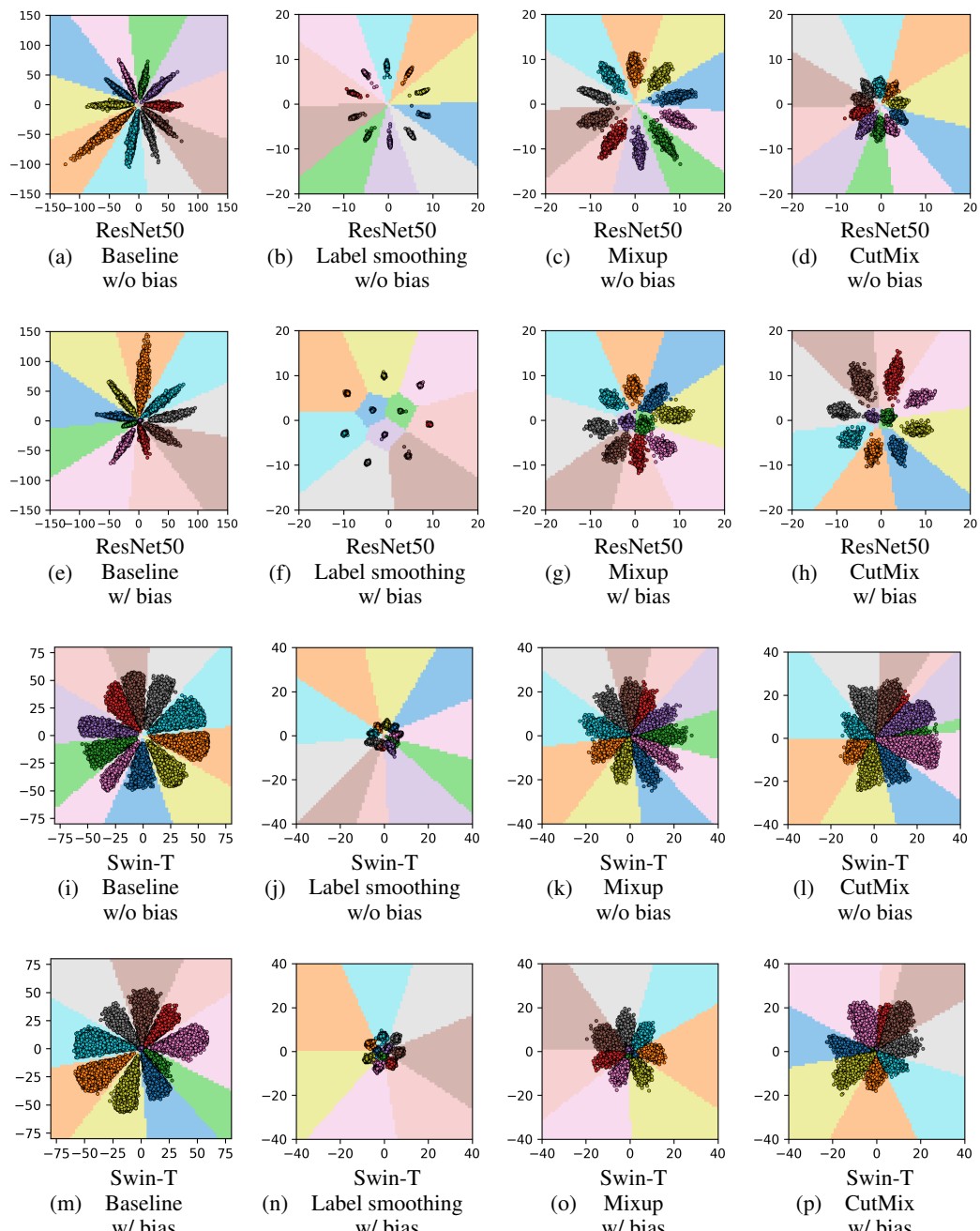

Figure 10: Decision regions and feature distribution in the 2D representation space for ResNet50 and Swin-T on CIFAR-10. Note that the scales differ across figures.

# E    DETERMINING CENTERS OF DECISION REGIONS

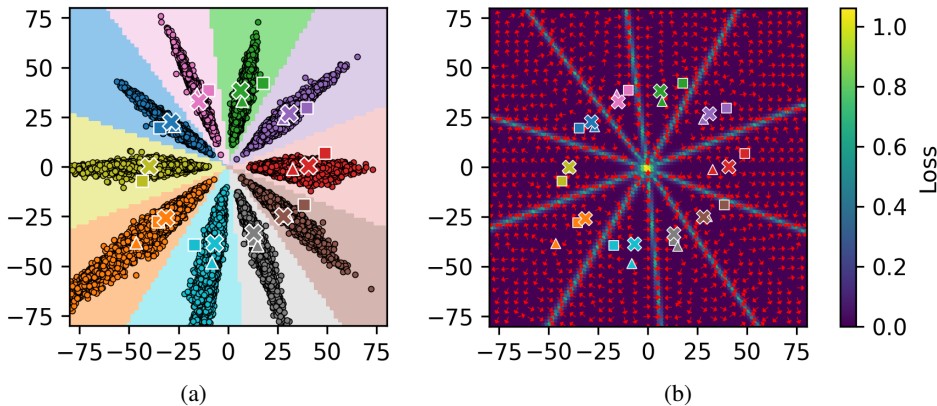

(a)                                            (b)

Figure 11: Class center candidates for ResNet50 with a 2D representation space on CIFAR-10. Triangles, squares, and crosses represent class means, weight vectors, and minimum loss points, respectively. (a) Features in the representation space. (b) Loss and gradient directions in the representation space.

It is natural to measure the proximity of a feature to the decision boundary based on its confidence. Following this idea, the cosine similarity of a feature with the *class center* should have the strongest correlation with its confidence, as the class center is expected to have the highest confidence (farthest from the decision boundary). In this section, we examine the following candidates for the class center: 1) the mean of correctly classified features within a class (*class mean*), 2) the *weight vector* of the classification layer, and 3) the point where the classification loss is the lowest (*minimum loss point*).

Figure 11 illustrates the positions of these class center candidates in the 2D representation space. For class means (triangles), while their positions seem reasonable, large errors could occur if outliers exist far from the feature clusters. As seen in the pink and brown classes, the weight vector (square) often shows a significant error, making it unsuitable as the center of the decision region. The minimum loss points (crosses) appear to best represent the class center. More quantitative analysis is provided in Table 5, where the cosine similarity between features and minimum loss points shows the highest correlation with confidence.

Table 5: Pearson correlation coefficient between confidence and cosine similarity of features to class means, weight vectors, and minimum loss points. The case with the highest correlation among the three candidates is marked in bold. Training details can be found in Appendix A.2.

| Model | Method | Weight Vector | Class Mean | Minimum Loss Point |
|---|---|---|---|---|
| ResNet50 | Baseline | 0.35 | 0.52 | **0.56** |
| | Label smoothing | 0.36 | 0.61 | **0.70** |
| | Mixup | 0.36 | 0.60 | **0.75** |
| | CutMix | 0.39 | 0.52 | **0.65** |
| Swin-T | Baseline | 0.41 | 0.55 | **0.56** |
| | Label smoothing | 0.56 | **0.64** | **0.64** |
| | Mixup | 0.48 | 0.63 | **0.64** |
| | CutMix | 0.48 | **0.58** | **0.58** |
| MobileNetV2 | PyTorch V1 | 0.44 | 0.60 | **0.61** |
| | PyTorch V2 | 0.44 | **0.60** | **0.60** |
| EfficientNet-B1 | PyTorch V1 | 0.48 | 0.59 | **0.60** |
| | PyTorch V2 | 0.29 | **0.67** | **0.67** |
| ViT-B/16 | PyTorch Swag Linear V1 | 0.43 | 0.52 | **0.53** |
| | PyTorch V1 | 0.57 | **0.61** | **0.61** |

## F    MORE RESULTS ON THE EFFECT OF REGULARIZATION

Figures 12 and 13 show the changes in feature distribution due to regularization across various models. For Swin-T in Figure 12, we manually train the model using the settings described in Appendix A.2. For the models in Figure 13, we use pretrained weights from PyTorch. Further details can be found in Appendix A.2.

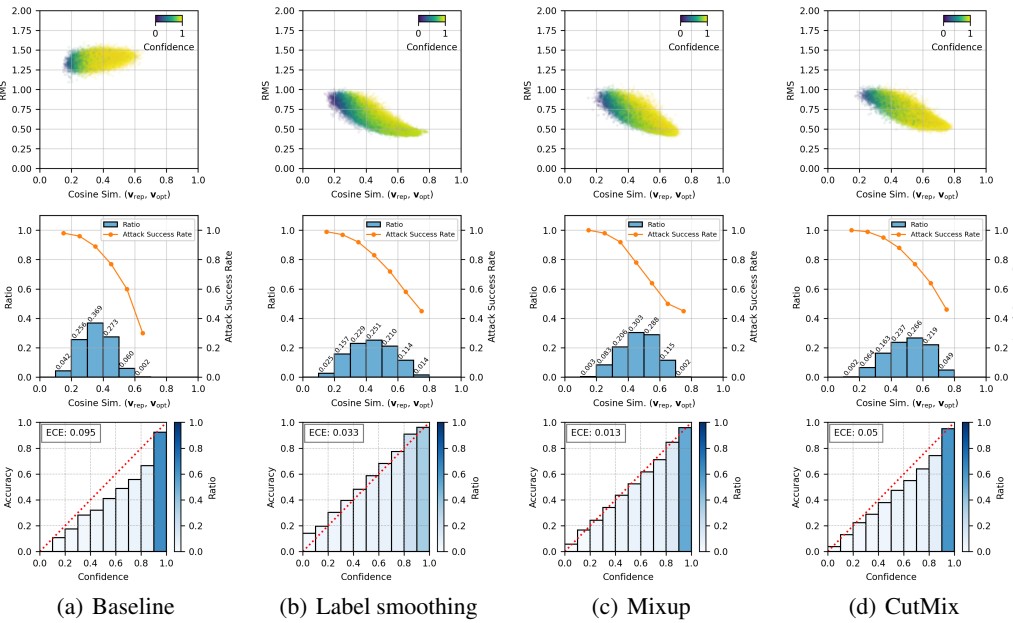

(a) Baseline     (b) Label smoothing     (c) Mixup     (d) CutMix

Figure 12: Evaluation results of Swin-T on the ImageNet validation data. **Top.** Scatter plots of feature RMS and cosine similarities of features with the class center. Colors represent confidence values. **Middle.** Histograms of cosine similarities of features to the class center, along with the FGSM attack success rate for each bin. **Bottom.** Reliability diagrams, where the transparency of bars represents the ratio of data in each confidence bin. ECE values are shown for each case.

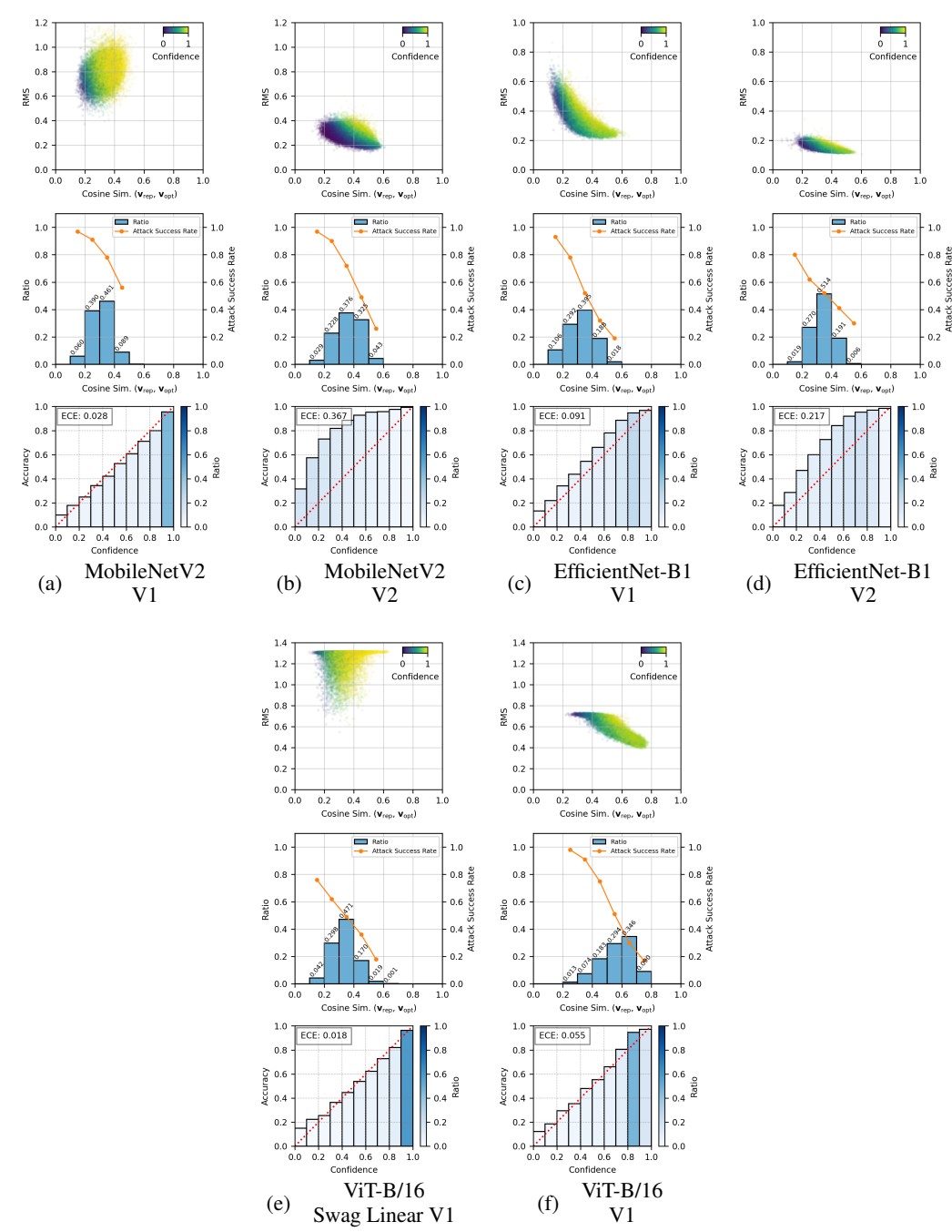

Figure 13: Evaluation results of MobileNetV2, EfficientNet-B1, and ViT-B/16 on the ImageNet validation data. **Top.** Scatter plots of feature RMS and cosine similarities of features with the class center. Colors represent confidence values. **Middle.** Histograms of cosine similarities of features to the class center, along with the FGSM attack success rate for each bin. **Bottom.** Reliability diagrams, where the transparency of bars represents the ratio of data in each confidence bin. ECE values are shown for each case.

# G  FEATURE SCALING

In Section 4.3, regularization using soft labels has an effect of scaling down of features. Here, we examine the possibility of manual feature scaling for calibration after training without regularization. In Figure 14, we show the accuracy and calibration performance of ResNet50 and Swin-T on the ImageNet validation set, before and after manually scaling features across various models and weights. $T = 1$ represents the use of original features. It can be observed that manual feature scaling does not affect classification accuracy but can improve calibration due to its similarity to temperature scaling.

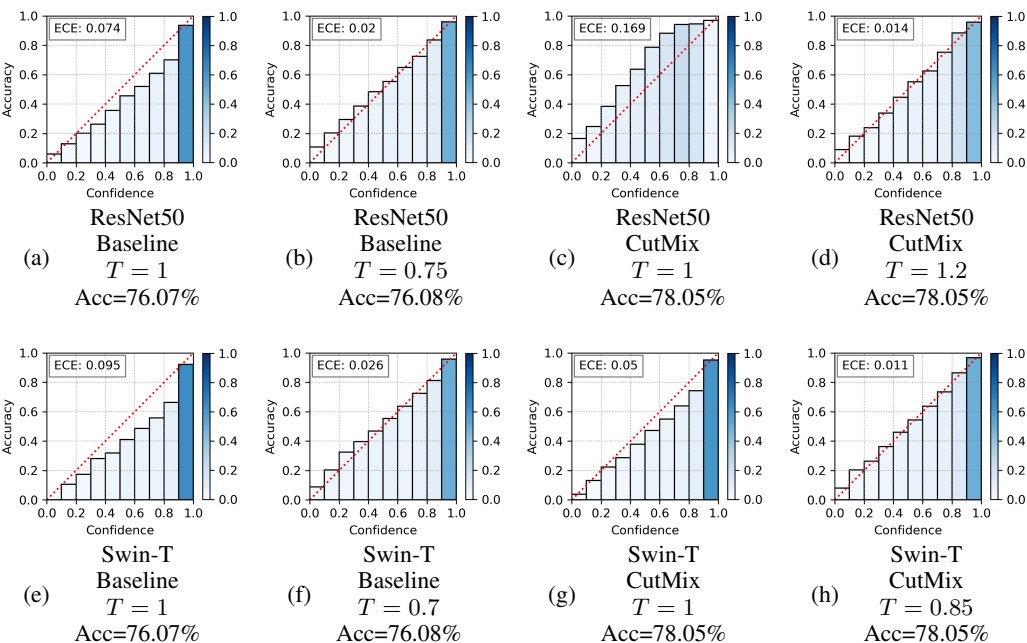

Figure 14: Calibration performances before and after manual feature scaling. $T$ represents the scaling factor, and Acc represents the accuracy on the ImageNet validation data.

## H  FEATURE RMS VS. PERTURBATION RMS

One may argue that regularized models could be expected to be also vulnerable to adversarial attacks because their features are close to the decision boundary near the origin. However, we find that the distance to the decision boundary near the origin is farther than that to the decision boundary at the side of the cone-shaped decision region when the distance is measured in the input domain. In Figure 15, we compare the feature RMS and perturbation RMS (i.e., the RMS of the difference between the features of clean and perturbed input images) for various models trained on ImageNet when the same amount of input perturbation ($\epsilon$=8/255) is applied. The positive correlation shown in the plot suggests that the features close to the origin change less than the features located far from the origin under the same amount of input perturbation. Therefore, robustness is determined mostly by the direction that a feature moves due to perturbation instead of the proximity of the feature to the decision boundary near the origin.

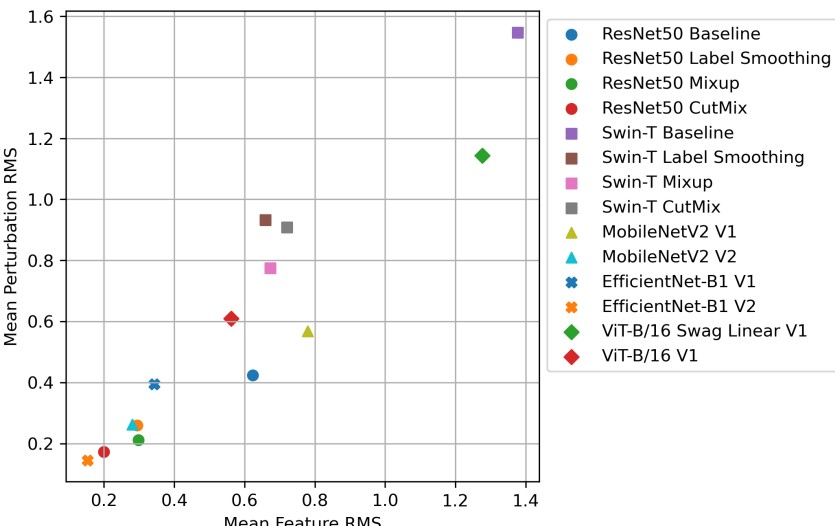

Figure 15: Scatter plot depicting the mean of feature RMS and perturbation RMS (the RMS of the difference between the features of clean and perturbed input images) across various models and training methods trained on ImageNet. Training details can be found in Appendix A.2.

