# OpenReview forum: "Impact of Regularization on Calibration and Robustness: From the Representation Space Perspective"
_ICLR.cc/2025/Conference — ICLR 2025 Conference Withdrawn Submission_

### Official Review · Reviewer_dznm · 2024-10-27

**Soundness:** 3
**Presentation:** 3
**Contribution:** 2
**Rating:** 5
**Confidence:** 4

**Summary:**

This manuscript is driven by the goal of improving overall model performance through the utilization of soft labels in various tasks, including classification, calibration, and robustness. The authors embark on a comprehensive exploration to analyze the underlying reasons for the observed performance enhancements associated with soft labels.

The work begins by providing empirical and theoretical insights into the structure of the embedding space learned by classification models. Notably, the authors demonstrate that this learned embedding space often exhibits a cone-like shape. This observation serves as a foundational element for their analysis, shedding light on the geometric properties of the representation space.

Building upon this understanding, the authors investigate various regularization techniques, including label smoothing, Mixup, and CutMix. They argue that these methods effectively promote tighter clusters among samples in the embedding space by reducing the Root Mean Square (RMS) of the feature representations. This reduction in RMS leads to enhanced model calibration performance and improved robustness, as the manuscript substantiates.

Through this analysis, the authors successfully connect the theoretical framework to practical improvements in model behavior, providing a compelling argument for the benefits of soft labels. The study contributes valuable insights into how regularization techniques can shape the embedding space and, consequently, enhance model performance across multiple dimensions.

**Strengths:**

1. The manuscript is well-written and thoughtfully structured. The authors effectively begin with a clear motivation that sets the stage for their research, articulating the central question they seek to address: understanding the origins of performance improvements attributed to soft labels. Following this introduction, they skillfully elaborate on their analytical framework for examining the decision regions within the representation space. Their findings culminate in the conclusion that these decision regions exhibit a cone-shaped structure.

2. The supportive evidence presented throughout the manuscript is both clear and compelling. Given the analytical nature of this work, the authors successfully navigate the lack of distinct boundaries between the methods and experimental sections. They seamlessly integrate detailed experimental results into the main text, ensuring clarity and continuity. Additionally, the figures are well-designed, effectively conveying complex information in a manner that is easy for readers to comprehend.

3. The principal concern this manuscript seeks to address relates to its significance in the broader context of machine learning research. While the authors present valuable insights into the mechanics of soft labels as a training technique, it is important to acknowledge that soft labels, despite being recognized as a general training trick, remain under-explored. This work opens up avenues for further investigation, specifically into the fundamental reasons behind their effectiveness. Expanding on this point could enhance the importance of their findings within the literature.

**Weaknesses:**

1. Trivial Conclusion Regarding the Cone-Shaped Structure:
 The authors arrive at the conclusion that the embedding space in classification tasks exhibits a cone-shaped structure. However, this finding seems somewhat trivial. The primary objective in classification tasks is to minimize the cross-entropy loss, which inherently means striving for a balanced decision boundary among classes. Given this context, it is to be expected that the learned class clusters will form cohesive groups in the embedding space. This assumption is not novel; it has been extensively utilized in the design of contrastive learning methods, including works like "Unsupervised Feature Learning via Non-Parametric Instance Discrimination" and "VICReg: Variance-Invariance-Covariance Regularization for Self-Supervised Learning." Therefore, the significance of the cone-shaped structure as a finding is diminished, as it does not provide new insights into the understanding of classification embeddings.

2. Weak Relationship Between Cone Shape and Subsequent Analysis:
 In Section 4.1, the authors aim to measure feature distributions using two criteria: RMS and cosine similarity. However, the manuscript lacks a clear articulation of how these criteria relate to the observed cone-shaped embedding structure. Despite devoting a considerable section (Section 3) to this topic, the authors should enhance their explanation to establish a more robust connection between the structure of the embedding space and the chosen evaluation metrics. Clarifying this relationship would enrich the analysis and provide a more comprehensive understanding of the significance of the cone shape in the context of the features being utilized.

3. Lack of Clear Insights:
 While the primary contribution of this manuscript is its thorough analysis, it fails to offer clear insights that could enlighten future research directions. For instance, the conclusion presented in Contribution 3, located in L066-P2, states that "feature vectors with smaller magnitudes improve model calibration and robustness." This assertion comes across as somewhat trivial within the broader context of machine learning. Historically, the aim has been to design compact features to achieve better classification performance. Techniques such as Principal Component Analysis (PCA)—while specifically targeting dimensionality reduction—essentially seek to form a more compact feature space. Thus, the manuscript would benefit from more profound insights or suggestions for future work that extend beyond such well-established concepts.

**Questions:**

See above

---

### Official Review · Reviewer_4JNV · 2024-11-01

**Soundness:** 2
**Presentation:** 2
**Contribution:** 2
**Rating:** 3
**Confidence:** 4

**Summary:**

This paper investigates the reasons behind the success of soft label approaches. It demonstrates that when using a linear classifier on top of a feature extractor, the learned representation space forms a cone shape in both 2D/3D and higher-dimensional spaces. Building on this discovery, the authors propose two novel metrics: the root mean square (RMS) of features and their cosine similarity to the class center vector. Through these metrics, the paper explains the effectiveness of soft-label approaches and identifies applications in calibration and model robustness.

**Strengths:**

This paper presents an interesting finding, showing that the representation space is cone-shaped when employing a linear classification head. Based on this finding, it introduces two novel metrics and offers a new perspective on a set of soft-label approaches.

**Weaknesses:**

Although this paper addresses an interesting problem, the current version lacks in-depth analysis. Concerns are expressed from the following perspectives:

1.	Limited Scope of Analysis: The analysis focuses solely on linear heads. The cone-shaped representation distribution appears primarily due to the linear head used with the feature extractor. However, many applications employ non-linear classifiers (e.g., MLP) or no classifier at all (e.g., in self-supervised learning). The existing analysis, therefore, narrows the applicability of the findings.

2.	Absence of Theoretical Analysis: While label smoothing, Mixup, and CutMix can all be viewed as utilizing soft labels, providing theoretical insights into these methods would enhance the paper's credibility.

3.	Lack of Proposed Methodology: Given that RMS and cosine similarity of features are identified as key factors, rather than merely analyzing existing methods through these lenses, the authors could propose new methodologies based on their observations.

4.	Weak Experimental Correlations: The connections between RMS, cosine similarity, and attack success rates are unclear, which undermines the conclusions drawn.

**Questions:**

I am unclear about why the representation space is cone-shaped if the first misclassified step is large. Could you provide a detailed explanation?

---

### Official Review · Reviewer_ZJsK · 2024-11-03

**Soundness:** 2
**Presentation:** 3
**Contribution:** 3
**Rating:** 5
**Confidence:** 4

**Summary:**

This paper investigates how regularization techniques using soft labels (like label smoothing, Mixup, and CutMix) effect the features space of image classification models. The key finding is that in standard image classifiers (both CNNs and Transformers) trained using hard labels, the decision regions in the representation space (features from the penultimate layer) form cone-like shapes around the origin. The features in these models often have high magnitude, which results in over-confident predictions. Regularization reduces the magnitude (RMS) of feature vectors and features become more tightly clustered around their class centers, resulting in higher cosine similarity between features and class centers. This results in improve model calibration by reducing overconfident predictions, which are caused by high magnitude features and improves robustness against adversarial attacks, due to tighter clustering. The authors use 2D/3D visualizations to demonstrate their analysis of the feature space. Multiple models are examined, including, ResNet50, MobileNetV2, Swin-T, and DeiT-T. The findings also suggest that the higher robustness of Transformers relative to CNNs might be partly explained by better regularization.

**Strengths:**

+ The topic and goal of the paper is laudable, improved understanding of fundamental aspects of training pipelines such as regularization is very important to making scientific progress.
+ The analysis is properly motivated and easy to follow
+ The visualizations stand out as exceptionally high quality and intuitively explain the analysis and findings.

**Weaknesses:**

- The number of experiments is insufficient, the authors discuss the impact of regularization on robustness, but only one set of experiments is carried out. Prior works study robustness much more thoroughly and also study different aspects of it, such as robustness to natural corruptions, robustness to black box attacks, etc.

- The claim about regularization improving calibration seems incorrect, if you look at Figure 5 a, b, c, and d, it appears the main effect of regulation is to reduce the confidence of the model. Since the baseline model is over-confident, reducing the confidence appears to improve calibration. However in the case of MixUp and CutMix it reduces the confidence so much that the model becomes poorly calibrated again, but in the direction of underconfidence. I suggest that this claim should be revised to state that it reduces the model’s confidence, which might not always result in a better calibrated model.

- Some of the claims in the paper are presented as novel findings, however these are well known in the community and among practitioners. Some instances of this are as follows:

- The claim that decision regions for cross-entropy trained models are cone-shaped is well known. In fact, prior works such as this technical report by [a] which compares different loss functions for person re-id uses a cone shaped region to represent the decision region even in a schematic figure.

- The point about robustness of CNNs vs Vision Transformers has been previously extensively studied. For instance refer to Pinto et.al [b] which presents a much more thorough examination of the question of CNN vs Transformers Robustness.

- Features becoming more tightly clustered around their class centers leading to higher robustness has been analyzed in Gupta et. al. [c] in the context of robustness of contrastive models.

I suggest that the authors acknowledge these prior works and clarify how their analysis extends the existing findings.


[a] Luo, Hao, Wei Jiang, Youzhi Gu, Fuxu Liu, Xingyu Liao, Shenqi Lai, and Jianyang Gu. "A strong baseline and batch normalization neck for deep person re-identification." IEEE Transactions on Multimedia 22, no. 10 (2019): 2597-2609.

[b] Pinto, Francesco, Philip HS Torr, and Puneet K. Dokania. "An impartial take to the cnn vs transformer robustness contest." In European Conference on Computer Vision, pp. 466-480. Cham: Springer Nature Switzerland, 2022.

[c] Gupta, Rohit, Naveed Akhtar, Ajmal Mian, and Mubarak Shah. "Contrastive self-supervised learning leads to higher adversarial susceptibility." In Proceedings of the AAAI Conference on Artificial Intelligence, vol. 37, no. 12, pp. 14838-14846. 2023.

**Questions:**

1. I would like the authors to respond to my point about the calibration claim. As mentioned above, it appears the regularization just makes the model less confident, not better calibrated.

2. It would be better to have a more thorough analysis of robustness. (See Weakness section for more details)

---

### Official Review · Reviewer_a7SU · 2024-11-04

**Soundness:** 4
**Presentation:** 4
**Contribution:** 4
**Rating:** 6
**Confidence:** 3

**Summary:**

Updating

**Strengths:**

Updating

**Weaknesses:**

Updating

**Questions:**

Updating

**Details Of Ethics Concerns:**

Updating

---

### Note · Authors · 2024-11-14

I have read and agree with the venue's withdrawal policy on behalf of myself and my co-authors.